# Lymphangioleiomyomatosis with Tuberous Sclerosis Complex—A Case Study

**DOI:** 10.3390/jpm13111598

**Published:** 2023-11-12

**Authors:** Aleksandra Marciniak, Jolanta Nawrocka-Rutkowska, Agnieszka Brodowska, Andrzej Starczewski, Iwona Szydłowska

**Affiliations:** Department of Gynecology, Endocrinology and Gynecological Oncology, Pomeranian Medical University in Szczecin, 71-252 Szczecin, Poland; jolanaw@poczta.onet.pl (J.N.-R.); agabrod@wp.pl (A.B.); andrzejstarcz@tlen.pl (A.S.); iwonaszyd@wp.pl (I.S.)

**Keywords:** lymphangioleiomyomatosis, tuberous sclerosis complex, benign metastasizing leiomyomatosis

## Abstract

Lymphangioleiomyomatosis (LAM) is characterized by lung cysts that cause lung deterioration, changes in the lymphatic system, and tumors in the kidneys. It mainly affects women of reproductive age and is a progressive disease. LAM can occur as an isolated disease or coexist with tuberous sclerosis (TSC). The source of LAM cells is unknown. Patients with confirmed LAM should be treated with an mTOR inhibitor, sirolimus, or everolimus. We present a case of LAM with TSC in a patient whose symptoms, including those in the lymph nodes and chyaloperitoneum, mainly concern the abdominal cavity.

## 1. Introduction

Lymphangioleiomyomatosis (LAM) is characterized by lung cysts that cause lung deterioration, changes to the lymphatic system, and tumors in the kidneys. It mainly affects women of reproductive age and is a progressive disease [1,2]. LAM may occur as an isolated disease or coexist with tuberous sclerosis (TSC) [1,3]. The incidence of sporadic lymphangioleiomyomatosis (S-LAM) is unknown, but one study found that the sporadic form occurs in 1 in 400,000 adult women [4]. The sporadic form of LAM (S-LAM) is due to an acquired mutation, mainly in the TSC2 gene [5,6]. These are cases with no other TSC symptoms. Unlike S-LAM, which mainly affects women, tuberous sclerosis complex lymphangioleiomyomatosis (TSC-LAM) affects both men and women with TSC [4]. The hereditary form of LAM (TSC-LAM) affects about 40% of patients with tuberous sclerosis (TSC). Most women with TSC develop LAM. TSC is a genetic disease of variable penetrance associated with many benign and rarely malignant tumors of the skin, eyes, brain, heart, lungs, liver, and kidneys. Mutations in the tumor-suppressor genes TSC1 and TSC2 lead to aberrant signaling through the mTOR pathway and are found in both S-LAM and TSC-LAM. These pathways participate in regulating cellular functions including growth, motility, and survival [7].

The essence of LAM is the presence of cysts in the lungs, which lead to a gradual loss of organ function. Cysts consist of abnormal muscle cells that enter the lungs through lymphatic vessels. The source of LAM cells is unknown. They migrate to the lungs, forming cyst walls and nodular changes. Histologically, they are spindle-shaped cells with a large amount of eosinophilic cytoplasm. LAM cells also produce large amounts of matrix metalloproteinases (MMPs). They are responsible for tissue remodeling and extracellular matrix degradation, participating in tumor migration, invasion, and metastasis. LAM cells stain positive for smooth muscle actin, vimentin, desmin, estrogen, and progesterone receptors and the monoclonal antibody human melanoma black (HMB-45). HMB-45 staining is particularly useful in distinguishing LAM from other lung lesions with smooth muscle predominance [1,5,7,8]. LAM cells also express lymphangiogenic growth factors (vascular endothelial growth factor (VEGF)-C and VEGF-D), which facilitate the metastatic spread of LAM cells and the breakdown of the extracellular matrix by matrix metalloproteinases (MMPs), implicated in cyst formation [2,8].

During periods of high hormone concentrations, i.e., the reproductive period, lung function declines due to estrogen and progesterone receptors in LAM cells. Estrogen stimulates cell proliferation and migration [1,5].

LAM cells carry inactivating mutations in the tumor-suppressor genes TSC1 and TSC2. These proteins normally limit cell growth by suppressing the mammalian target of rapamycin complex 1.

The common early symptoms of LAM include shortness of breath, pneumothorax, and chest pain. The main clinical manifestations of LAM are dyspnea, pneumothorax (30% of patients); pleural effusion (10–30% of patients), mainly chylothorax; renal AML (30% of S-LAM patients and 80% of TSC-LAM patients); lymphangioleiomyoma (16–38% of patients); and, less frequently, chyloperitoneum, chyluria, and chylopericardium [2,9,10,11]. 

The diagnostic method of choice is high-resolution computed tomography (HRCT) of the lungs. HRCT is very sensitive and specific in detecting LAM and can reveal characteristic diagnostic findings of numerous thin-walled pulmonary cysts. Other radiographic findings may include reticulonodular opacities, pleural thickening, hyperinflation, and focal ground-glass opacities [4].

Seventy percent of LAM patients show elevated levels of VEGF-D, a growth factor that binds to VEGF receptor 3. Current guidelines recommend measuring serum VEGF-D in all patients with suspected LAM. In doubtful cases, a lung biopsy may be considered [5]. In order to assess the pathology of the abdominal cavity, CT or magnetic resonance imaging is performed [2,10].

In 2010, the European Respiratory Society presented guidelines for the diagnosis of LAM. Based on lung HRCT, characteristic CT features, and/or on pathological examination, visual and/or biochemical characteristics of the effusion and clinical history the diagnosis of LAM is made as definite LAM, probable LAM, or possible LAM [11].

Patients with confirmed LAM should be treated with an mTOR inhibitor, sirolimus, or everolimus [2,11,12,13].

LAM occurs almost exclusively in women of reproductive age, and the presence of estrogen and progesterone receptors in LAM cells suggests that hormones influence the development and progression of the disease. Disease progression occurs during periods with high estrogen concentrations, e.g., during pregnancy or hormonal therapy, while the development of symptoms declines after menopause. Some studies suggest that anti-estrogen therapy could be a recognized treatment of LAM [1,7,14]. Reports on treatment with gestagens and estrogen receptor modulators are unclear and require further research. Despite a number of case reports describing the use of oophorectomy, tamoxifen, and GnRH agonists in LAM, there are no confident data on the efficacy of any antiestrogen strategy for LAM. Recommendations indicate that hormone treatments other than progesterone should not be used in patients with LAM [11]. There is also no evidence for the effectiveness of treatment with doxycycline in LAM [12].

## 2. Case Presentation

In June 2020, a 38-year-old patient was admitted to the Department of Gynecology, Endocrinology, and Gynecological Oncology of Pomeranian Medical University in Szczecin due to abdominal pain. She had her first period at age 15 and her last period at age 38. She had irregular and heavy menstrual bleeding. She was never pregnant. During hospitalization, CT of the abdominal cavity was performed and the paraaortic lymph nodes were found to be enlarged up to 15 mm, and arranged in bundles. The right iliac lymph nodes were enlarged to 21 mm and the left to 22 mm. Mesenteric lymph nodes were enlarged to 9 mm. In addition, in the kidneys, one of the foci had a 7 mm diameter of fat density, which implies AML. 

The patient previously underwent two laparotomies with partial removal of the uterine fundus due to heavy menstrual bleeding, in 2009 and 2012. The obtained histopathological results revealed adenomyosis. In 2017, due to persistent vaginal bleeding, the patient underwent removal of the uterine corpus without an adnexa. In 2016, she underwent another laparotomy due to lymphoma suspicions. At that time, CT of the abdominal cavity revealed significantly enlarged lymph nodes. Histopathological examination diagnosed metastatic myoma. This examination revealed no atypia, necrosis, or mitotic activity. IHC: CD 34+, CD31+, SMA+, desmin+, and MIB-1+ in singe cells. 

The June 2020 hospitalization was prompted by a significant progression of the lymph node dimensions and suspicions of malignant transformation into sarcoma. Therefore, the patient qualified for laparoscopy iliac lymph node collection. On admission, all laboratory tests were normal. The histopathological results confirmed a benign myomatosis diagnosis. The tumor’s receptor status was also determined. Immunohistochemical staining revealed cells positive for estrogen and progesterone receptors (IHC: ER ++ in 60% cells, PR +++ in 60% cells).

A month after the operation, the patient was admitted to the hospital again due to a growing abdominal circumference and massive ascites. An abdominal ultrasound revealed ascites and a chest X-ray revealed a small amount of fluid in the pleural cavities. In laboratory tests, no deviations were found, apart from the reduced concentrations of total protein and albumin, and elevated concentrations of CA 125 and HE4 markers. An abdominal puncture was performed, and 1000 mL of fluid was collected for cytological assessment. The result was characteristic of exudate. The obtained fluid (lymph) accumulated in the abdominal cavity after lymph node extraction. Due to persistent ascites, with clinical symptoms (dyspnea and pain), ascites puncture was repeated after a month and again after 6 weeks of hospitalization, with 1500–2000 mL of fluid (lymph) obtained each time. 

The patient’s medical history indicates that she has epilepsy and was diagnosed with tuberous sclerosis—Bourneville’s disease. The patient met the diagnostic criteria for TSC, which includes cutaneous manifestations (hypopigmented macules on the skin of the trunk and limbs), neurological manifestations (nodular lesions in the brain characteristic of TSC as described by MRI and epilepsy), and renal manifestations (angiomyolipoma (AML)). A genetic oncology clinic diagnosed the patient with tuberous sclerosis complex (TSC) based on clinical symptoms in 2021.

Due to the patient’s BML (benign metastasizing leiomyomatosis) diagnosis with (+) ER and (+) PR and severe clinical symptoms (disease progression), we conducted a multidisciplinary analysis of this case with a rheumatologist and an oncologist. The medical council decided on a hormonal treatment with long-acting GnRH analogues (aGnRH) and a selective estrogen receptor modulator—Tamoxifen. The patient’s clinical condition improved as a result of our treatment. 

In June 2021, the patient was admitted to the Department of Endocrinology due to suspicions of a pancreatic neuroendocrine tumor. Based on laboratory tests, a pancreatic neuroendocrine tumor was excluded. A CT scan of the abdominal cavity and pelvis showed progression in the size of the iliac and periaortic lymph nodes and an increase in the abdominal fluid. After multiple abdominal ascites punctures, we decided to persistently drain the peritoneal cavity for one week to avoid repeated punctures.

Chest CT revealed numerous disseminated, thin-walled cysts in the lungs, small nodules of various densities, and a 6 mm “milk glass” nodule (Figure 1).

The final LAM diagnosis was determined by the team and thoracic surgeons based on clinical condition analysis and radiological examination (including HRCT). Treatment with mTOR inhibitors (everolimus or sirolimus) was recommended due to predominant abdominal symptoms. At the beginning, this therapy was combined with long-acting aGnRH and aromatase inhibitors. After clinically improving all symptoms, resolving ascites, and reducing lymph nodes size in the control CT examination, we decided to discontinue the aGnRH therapy to avoid adverse side effects. The follow-up revealed no recurrence of symptoms and good patient overall well-being for 14 months. We also decided to discontinue aromatase inhibitor therapy.

The recommended concentration of sirolimus in serum (5–15 ng/mL) was maintained at a dose of 1–2 mg per day. Additionally, the patient was treated for epilepsy with lamotrigine by a neurologist.

## 3. Discussion

The pathogenesis of LAM is unclear. LAM can be divided into sporadic and tuberous sclerosis-associated forms. Both forms of the disease can cause pulmonary symptoms. Extrapulmonary LAM has the same morphological features as pulmonary LAM and may occur concomitantly with or precede it [15]. In patients with LAM, it is characteristic that lung function deteriorates [1,2,12]. In the presented case, the patient’s symptoms were chyaloperitoneum and the presence of enlarged lymph nodes in the abdominal cavity. She had no typical pulmonary symptoms. Such a rare cases of extrapulmonary lymphangioleiomyoma (E-LAM) was described earlier in the literature [16,17,18,19].

Morphologic differential diagnoses include metastatic leiomyomas, metastatic leiomyosarcoma, gastrointestinal stromal tumor (GIST), and malignant melanoma [20]. Bourneville’s disease is diagnosed based on genetic tests (mutations in the TSC1 or TSC2 genes) or by assessing clinical symptoms of the disease. The most important symptoms are skin angiofibromas, skin spots, multiple retinal hamartomas, cortical brain nodules, gigantum astrocytoma, heart rhabdomyosarcoma, lung lymphangiloleiomyatosis, and renal angiolipoma. In addition, patients often have epilepsy and intellectual disabilities or behavioral disorders [21]. Pulmonary manifestations of TSC include lymphangioleiomyomatosis (LAM), which is seen in about 30–40% of reproductive-age women [1]. Our patient was diagnosed with tuberous sclerosis alongside epilepsy, a neurological manifestation, and angiomyolipoma in the kidneys. Imaging examinations of the lungs showed characteristic for LAM CT image [1,2,12]. She had numerous tumors in the lymph vessels in the abdominal cavity. This kind of myomatosis is what causes the main troublesome clinical symptoms. It is difficult to determine whether this is a separate symptom or a variant that may also contribute to tuberous sclerosis.

The dysregulation of the mTOR signaling pathway is the main cause of abnormal LAM cell proliferation [22]. Microscopically detectable smooth muscle cell proliferation is the main cause of bronchiole obstruction and alveolar air blockage leading to spontaneous pneumothorax. In addition, blockage of the lymphatic vessels leads to chylothorax and chylic ascites [23,24,25]. Research on extrapulmonary LAM treatment has been limited [16,17,18,19]. Since LAM occurs in approximately 30% of women with TSC alongside TSC1 and TSC2 tuberous sclerosis gene mutations, TSC genes play an important role in cell cycle regulation through the mammalian target rapamycin (mTOR) signaling pathway. Sirolimus, also called rapamycin (an mTOR inhibitor), blocks the mTOR signaling pathway and restores homeostasis in LAM cells [3,26]. Treatment with mTOR inhibitors results in significant improvement, primarily by resolving the chyloperitoneum. It also improves lung function and the quality of life for people with LAM. For this reason, after clinical trials, the Food and Drug Administration (FDA) announced its first-ever approval of sirolimus as a treatment for LAM in 2015 [27,28,29,30].

LAM cells express estrogen and progesterone receptors, and lung function declines when circulating estrogen levels are high. Many studies have shown that estrogen is an important factor in LAM cell proliferation, migration, and metastasis. Due to the coexpression of smooth muscle proteins (actin and desmin) and melanocyte markers (HMB-45, Melan-A, and MART-1), LAM cells are thought to derive from perivascular epithelial cells, although this is still unclear [1,9,14].

A pathology that should not be confused with lymphangioleiomyomatosis (LAM) is BML. In the presented patient, the histological result of the lymph nodes revealed metastatic myoma with cells positive for estrogen and progesterone receptors. A lack of common pulmonary symptoms of LAM delayed the patient’s diagnosis. In the literature, it is highlighted that E-LAM is difficult to diagnose [16,17,18,19].

Unlike BML, LAM causes atypical smooth muscle cell proliferation along blood vessels, lymph vessels, and small airways. HMB-45 immunohistochemical staining is positive for LAM but negative for BML [25].

Currently, assessing the concentrations of the lymphatic growth factors VEGF-C and VEGF-D helps in the diagnosis of LAM without requiring a lung biopsy. The fact that serum VEGF-D is elevated in 70% of LAM patients makes it a clinically useful diagnostic and prognostic biomarker. In addition, sirolimus, a drug that inhibits VEGF-D function and lymphocyte proliferation, stabilizes lung function and minimizes complications in LAM cases [13]. Recent research suggests a novel biomarker, FGF23 (a thirty-two-kDa protein secreted by osteocytes), to measure LAM activity. The authors assessed FGF23’s association with pulmonary diffusion abnormalities in LAM patients [3].

New strategies of LAM treatment will probably be based on LAM cell-killing therapy in combination with sirolimus. Currently recommended therapies do not include an effective antiestrogen hormonal treatment. It is possible that, in the future, clinical trials will show a beneficial therapeutic effect of other hormonal therapeutic approaches in patients with LAM [11,12]. Estrogen-reducing nutrition may also be strategy for patients with LAM.

This case suggests that personalized treatment strategies should be implemented for every patient with LAM. Differentiating LAM from BML is also important at the beginning of diagnosis. This study will be helpful in properly and effectively treating patients.

## Figures and Tables

**Figure 1 jpm-13-01598-f001:**
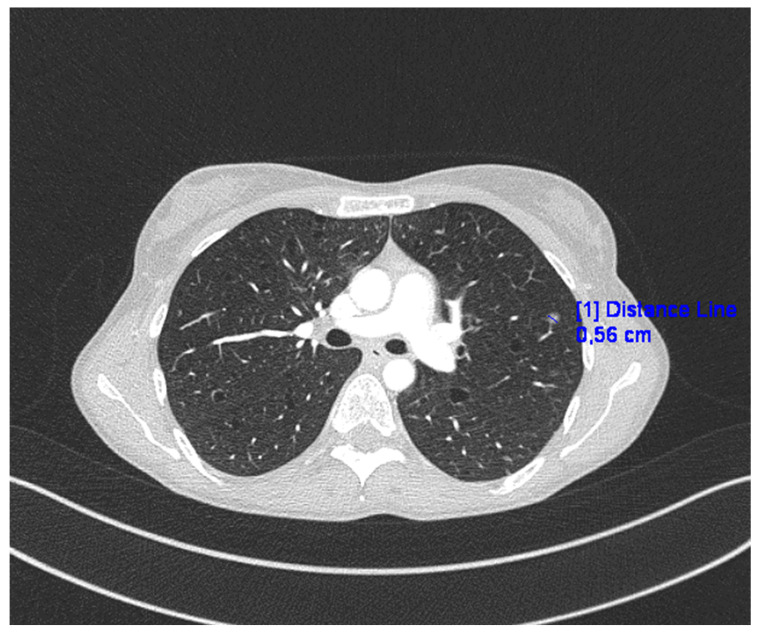
CT of the lungs in presented case.

## Data Availability

The data presented in this study are available upon request from the corresponding author.

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
