# Peer review of "Lymphangioleiomyomatosis with Tuberous Sclerosis Complex—A Case Study"

_jpm, 2023, doi:10.3390/jpm13111598_

Round 1

Reviewer 1 Report

Comments and Suggestions for Authors

Introduction is inconsistent and there are parts that are repeating in a little bit different way. I suggest putting epidemiology first, then pathogenesis, clinical picture and manifestation, diagnostic procedures and the treatment. There is no references on LAM guidelines for diagnosis and management published by ERS in 2010 and ATS and Japanese Thoracic Society in 2016. In that guidelines there is no recommendation to treat LAM patients with hormonal therapy and doxycycline is not recommended which is not clearly said in this paper. The main lung manifestations are thin-walled cysts and in this article lung tumours are mentioned which can lead to confusion because there is abnormal proliferation od smooth muscle cells in bronchial wall and lymph vessels and the consequence is formation od cysts.

Case presentation should have more details about last admission- clinical data, laboratory findings, pregnancies and it would be very nice to have some radiologic pictures of CT scan. I have to say that we can’t say that CT scan lung finding was non-specific because thin-walled cysts were described which can be one of diagnostic criteria for LAM. Instead of consulting thoracic surgeon, pulmonologist and thoracic radiologist should be consulted to be sure if these changes are significant or not. Since we have clear guidelines how to treat LAM, it is not clear why is this patient treated with letrozole.

I also do not agree about differential diagnosis mentioned in discussion especially when we talk about lung manifestations of LAM. Then sarcoidosis is not a differential diagnosis. It is also mentioned that this patient didn’t have typical tumor locations within lungs and this is confusing because we do not expect tumors, but cysts in lungs.

I suggest to do the revision of chest CT finding with thoracic radiologist and to comment diagnostic and treatment procedures according to current guidelines and what was the reason of delayed diagnosis in patient who already had diagnosis of tuberous sclerosis.

Author Response

Dear Sir or Madame,

Thank you for taking the time to review our article and for all your helpful suggestions. All requested changes have been implemented in a revised version of the article and marked as red parts in the text. We do hope our modifications will meet your expectations and standards. 

For your convenience, all points raised in the review are addressed below in separate paragraphs, point-by-point. 

Comments

1. According to the suggestions, the introduction has been changed. We modified the text order: epidemiology first, then pathogenesis, clinical picture and manifestation, diagnostic procedures and the treatment. We have included recommendations published by ERS in 2010 and ATS and Japanese Thoracic Society in 2016. We have changed the terminology to avoid any confusion in the assessment of lung pathology.

2. We also added data in the case report, including analyzed CT result and CT scans.

3. The patient was consulted with a thoracic surgeon, because in our clinical center this type of pathology is consulted by thoracic surgeons.

4. The patient was initially treated with letrozole due to histopathological result - benign metastasizing leiomyoma (BML). That delay the final diagnosis, but on the base of clinical picture and chest CT we diagnosed LAM and previous treatment was modified, according to reccomendations.

5. We have changed the discussion and description of the differential diagnosis as well as the terminology regarding the changes described in the patient's imaging tests.

6. During the observation patient's therapy resulted in the reduction of chyaloperitoneum and pain. Letrozole was administered before the diagnosis of LAM and was based on the first histopathological result: BML which the presence of estrogen and progesterone receptors. Continued treatment (sirolimus) is consistent with the recommendations of the ATS and Japanese Thoracic Society.

Once again, thank you for your time and useful suggestions – all are much appreciated. 

With kind regards,

Aleksandra Marciniak

Reviewer 2 Report

Comments and Suggestions for Authors

Thank you for the possibility to review this important article on hot topic:  Lymphangioleiomyomatosis (LAM) with Tuberous Sclerosis Complex - A Case Study

I have some proposals:

1.     Please include imaging and pathomorphological examinations results in your case presentation

2.     I think it will be nice if you will generalize multiple LAM manifestations in one table or figure

3.     In discussions please overview and compare your case with cases from the literature

4.     Please present future directions of study of LAM

Author Response

Dear Sir or Madame,

Thank you for taking the time to review our article and for all your helpful suggestions. All requested changes have been implemented in a revised version of the article and marked as red text. We do hope our modifications will meet your expectations and standards. 

For your convenience, all points raised in the review are addressed below in separate paragraphs, point-by-point. 

Comments

1. We included CT scans in our case report. The pathological images, on which we based the initial diagnosis suggested BML, not LAM. Our final diagnosis was based on clinical symptoms and chest CT scan. So we didn't attach a histopathological images to avoid misdiagnosis.
2. As we cited  in introduction section LAM guidelines for diagnosis and management published by ERS and ATS and Japanese Thoracic Society we decided not to include LAM symptoms in special table. In above guidelines LAM manifestations are clearly presented.
3. We added literature presenting especially E-LAM cases to the discussion, including a comparison to presented case.
4. According to the recommendations cited in the text, the current standard of treatment for LAM are mTOR inhibitors. But we also discussed future perspectives in LAM therapy, as suggested

Once again, thank you for your time and useful suggestions – all are much appreciated. 

With kind regards,

Aleksandra Marciniak

Round 2

Reviewer 1 Report

Comments and Suggestions for Authors

No further comments, authors accepted all previous comments and made suggested changes. 

Reviewer 2 Report

Comments and Suggestions for Authors

Thank you for your revision!